# Genetic and Comparative Genome Analysis of *Exiguobacterium aurantiacum* SW-20, a Petroleum-Degrading Bacteria with Salt Tolerance and Heavy Metal-Tolerance Isolated from Produced Water of Changqing Oilfield, China

**DOI:** 10.3390/microorganisms10010066

**Published:** 2021-12-29

**Authors:** Zhaoying Su, Shaojing Wang, Shicheng Yang, Yujun Yin, Yunke Cao, Guoqiang Li, Ting Ma

**Affiliations:** Key Laboratory of Molecular Microbiology and Technology, Ministry of Education, College of Life Sciences, Nankai University, Tianjin 300071, China; suejoing@163.com (Z.S.); 1120200495@mail.nankai.edu.cn (S.W.); 2120201130@mail.nankai.edu.cn (S.Y.); yujun_yin@163.com (Y.Y.); 2120201145@mail.nankai.edu.cn (Y.C.); gqli@nankai.edu.cn (G.L.)

**Keywords:** *E. aurantiacum* SW-20, comparative genomics, alkane degradation, heavy metal tolerance, salt-tolerance

## Abstract

The genome of *Exiguobacterium aurantiacum* SW-20 (*E. aurantiacum* SW-20), a salt-tolerant microorganism with petroleum hydrocarbon-degrading ability isolated from the Changqing Oilfield, was sequenced and analyzed. Genomic data mining even comparative transcriptomics revealed that some genes existed in SW-20 might be related to the salt tolerance. Besides, genes related to petroleum hydrocarbon degradation discovered in genomic clusters were also found in the genome, indicating that these genes have a certain potential in the bioremediation of petroleum pollutants. Multiple natural product biosynthesis gene clusters were detected, which was critical for survival in the extreme conditions. Transcriptomic studies revealed that some genes were significantly up-regulated as salinity increased, implying that these genes might be related to the salt tolerance of SW-20 when living in a high salt environment. In our study, gene clusters including salt tolerance, heavy metal tolerance and alkane degradation were all compared. When the same functional gene clusters from different strains, it was discovered that the gene composition differed. Comparative genomics and in-depth analysis provided insights into the physiological features and adaptation strategies of *E. aurantiacum* SW-20 in the oilfield environment. Our research increased the understanding of niches adaption of SW-20 at genomic level.

## 1. Introduction

*Exiguobacterium aurantiacum* (*E. aurantiacum*) is a genus of bacilli and a member of the low GC phyla of Firmicutes. Collins et al. first described the genus *Exiguobacterium* with the characterization of *E. aurantiacum* strain DSM6208T from an alkaline potato processing plant [1]. *Exiguobacterium* with a high tolerance of NaCl (12% NaCl) and could live in a wide range of temperature [2,3], indicated that different kinds of *Exiguobacterium* could be isolated from different temperatures. Because of their ability to survive in a variety of temperature extremes, they are an important area of study. Some strains, in addition to dynamic thermal adaptation, are also halotolerant (13% NaCl was added to the medium), can grow within a wide range of pH values (5–11), and tolerate high levels of UV radiation and heavy metal stress (including arsenic) [4]. Previous studies have shown that most members from *Exiguobacterium* genus isolated to date are Gram-positive and generally rod-shaped [5,6,7]. When in logarithmic or stable growth phases, the majority of them are facultative anaerobic, and some are spherical [8]. Colonies formed in the nutritional medium under aerobic conditions are mostly round and light yellow or even orange, whereas colonies formed under anaerobic conditions are milky white; to a certain extent, the change of pigment content can also reflect the growth characteristics of bacteria [9,10].

Previous research found *E. aurantiacum* to be an alkaliphilic extremophile [11]. Some extremophiles can be used in various applications due to the presence of several special enzymatic activities that allow them to gain resistance to xenobiotics or heavy metals and even have a good potential to produce exopolysaccharide or biosurfactants [12]. It is worth mentioning that *E. aurantiacum* has been reported to degrade aliphatic hydrocarbon, pyridine and even naphtaléne [13]. The completion genomes of two strains, *Exiguobacterium chiriqhucha* RW2 and *Exiguobacterium chiriqhucha* GIC31, were confirmed by Gutiérrez-Preciado et al. [14]. Phospholipid fatty acid analysis (PLFA) of *Exiguobacterium chiriqhucha* strain RW2 at different temperatures causes major changes in cellular membrane function, which may allow for its temperature, pH and salinity adaptation [14].

Here, the complete genome sequence of *E. aurantiacum* SW-20, which was isolated from the reservoir formation water from Changqing Oilfields (China), was present. The genomic size is 2,953,062 bp long, with a GC% content of 52.19%. Some genes involved in the degradation of petroleum hydrocarbon, salinity-tolerance and heavy metal tolerance were identified among the sequenced genes, indicating that this species has the potential to be used as an environmental decontaminating agent and for the bioremediation of extreme environments. Moreover, it may serve as a starting point for further research into salt-tolerant microorganisms in reservoir environments.

## 2. Materials and Methods

### 2.1. Isolation of E. aurantiacum SW-20

Samples of produced water were collected from the reservoir environment of Changqing Oilfields (China) and *E. aurantiacum* SW-20 was isolated using Luria-Bertani (LB) medium. The LB liquid medium consists of yeast powder 10 g, NaCl 5 g and peptone 5 g and ration to 1 L of distilled water. For the solid medium, an additional 1.5% agar powder was added on this basis. Specifically, to explore salt tolerance of 5% NaCl, based on the LB liquid medium, an additional 50 g/L of NaCl was added and then cultured for three days at 37 °C constant-temperature on shaking table in a 100 mL shake-flask. When the culture medium became cloudy, it was subcultured two to three times under the same conditions to ensure that the bacteria remain active. Gradient dilution and streaked culture on the same conditions in LB solid medium (with an additional 5% NaCl) were then performed. When independent colonies formed, some were selected and enriched in the same salt condition. The acclimation culture was then carried out by gradually increasing the salinity. When the concentration of NaCl was increased at 8%, only a few monoclonal colonies formed. Monoclonal colonies were selected with a sterile spear tip and 16S rRNA was sequenced using universal bacterial primers 27F and 1492R after culture. When independent colonies formed, selected some and enriched under the same salt conditions, then the acclimation culture was carried out by increasing the salinity step by step.

### 2.2. Phylogenetic Identification and Genome Sequencing

High quality genomic DNA of *E. aurantiacum* SW-20 was extracted using a Wizard^®^ Genomic DNA Purification Kit (Promega, Madison, WI, USA). DNA extracts were verified using gel electrophoresis on 1% agarose and 16S rRNA gene was amplified using the universal bacterial 16S primers [15]. The purified PCR products were sequenced using Illumina HiSeq X Ten and the 16S rDNA sequences of *E. aurantiacum* SW-20 and some type strains belonging to the same phylogenetic group were downloaded and compared to the NCBI database. The phylogenetic tree was constructed in MEGA V.11 using Maximum-likelihood methods of tree making algorithms [16].

A whole genome sequence was performed on an Illumina HiSeq X Ten platform. Briefly, DNA was sheared into 400–500 bp fragments using a Covaris M220 Focused Acoustic Shearer in accordance with the manufacture’s protocol. Illumina sequencing libraries were prepared from the sheared fragments using the NEXTflex™ Rapid DNA-Seq Kit. The prepared libraries were then used for paired-end Illumina sequencing (2 × 150 bp) on an Illumina HiSeq X Ten machine. The data generated by the Illumina platform were analyzed using bioinformatics. All the analyses were performed using I-Sanger Cloud Platform (www.i-sanger.com (accessed on 22 June 2021)) from Shanghai Majorbio. The detailed procedures are as follows: Firstly, the original image data was transferred into sequence data via base calling, which is defined as raw data or raw reads and saved as FASTQ file. Then, the FASTQ files of the original data were made available to users, which included detailed read sequences and read quality information. Finally, a statistic of quality information was applied for quality trimming so that low quality of data could be removed to form clean data. An assembly of the clean reads were performed using SOAPdenovo v.2.21 [17]. Raw reads were filtered using Trimmomatic v.0.39 [18] and then assembled by SPAdes v.3.13.1 [19] with default parameters. CDS and tRNA were predicted using Prokka v.1.13 [20] (http://trna.ucsc.edu/software/ (accessed on 22 June 2021)) and tRNA-scan-SE [21] respectively. Swiss-prot database [22] (https://web.expasy.org/docs/swiss-prot_guideline.html (accessed on 22 June 2021)), Pfam database [23] (http://pfam.xfam.org/ (accessed on 22 June 2021)), eggNOG database [24] (http://eggnogdb.embl.de/#/app/home (accessed on 22 June 2021)) and KEGG database [25] was performed for function annotation, rRNA was predicted using Barrnap (https://github.com/tseemann/barrnap/ (accessed on 22 June 2021)).

### 2.3. RNA-Seq Experiments of E. aurantiacum SW-20

Total RNA was extracted from the tissue using the TRIzol^®^ Reagent (Invitrogen, Waltham, MA, USA) according to the manufacturer’s instructions, and genomic DNA was removed using DNase I (TaKara, Kusatsu City, Japan). Next, RNA quality was determined by a 2100 Bioanalyser (Agilent, Santa Clara, CA, USA) and quantified using an ND-2000 (NanoDrop Technologies, Wilmington, DE, USA). Only high-quality RNA (OD 260/280 = 1.8–2.0, OD 260/230 ≥ 2.0, RIN ≥ 6.5, 28 S:18 S ≥ 1.0, ≥100 ng/μL, ≥2 μg) was used to construct sequencing library. RNA-seq library was prepared following TruSeqTM RNA sample preparation Kit from Illumina (San Diego, CA, USA). After being quantified by the TBS380, a paired-end RNA-seq sequencing library was sequenced with the Illumina HiSeq×TEN (2 × 150 bp read length). The data generated by the Illumina platform were analyzed using bioinformatics analysis. All analyses were carried out on the Majorbio Cloud Platform (www.majorbio.com (accessed on 5 September 2021)) from Shanghai Majorbio Bio-pharm Technology Co., Ltd. The major software and parameters are as follows. High quality reads in each sample were aligned to a genome using Bowtie2 v.2.3.5.1 [26]. Quantify gene and isoform abundances were calculated using RSEM v.1.3.3 [27]. Then differential expression analysis was performed by SARTools v.1.9 [28].

### 2.4. Characteristics of E. aurantiacum SW-20

Aside from salt-tolerance, their degradation of crude oil and the tolerance to heavy metal arsenic were also studied to better explore their characteristics. It is worth mentioning that the salt tolerance of *E. aurantiacum* SW-20 was mainly studied in LB medium. To explore the salt tolerance, we experimented with different NaCl concentrations in the LB medium, for example, 8% NaCl means add an additional 8% NaCl in the LB medium. A previous study has showed that *E. aurantiacum* could grow in the medium when take the hydrocarbon compounds (diesel and petrol) as the sole carbon source [29]. Crude oil degradation was mainly carried out in a crude oil degradation medium. The composition of crude oil degradation medium includes K_2_HPO_4_ 3H_2_O 4.8 g/L, KH_2_PO_4_ 1.5 g/L, (NH_4_)_2_SO_4_ 1.0 g/L, MgSO_4_ 7H_2_O 0.2 g/L, NaCl 60 g/L and yeast 0.01 g/L and 2% crude oil from Changqing oilfield was added as the sole carbon source. pH was adjusted to 7.2 and used after sterilization at 120 °C for 20 min. Petroleum hydrocarbons were extracted by solvent extraction method and analyzed via gas chromatography [3] when cultured in a 30 °C, 200 rpm shaker for seven days, and the morphology of SW-20 was also visualized through SEM analysis [30].

### 2.5. Comparative Genome and Genome Synteny Analyses

To better compare and analyze the genome of *E. aurantiacum* SW-20, four other strains from the same genus, including *E. mexicanum* A-EM (GCF_005960665), *E. mexicanum* HUD (GCF_000763125), *E. aurantiacum* DSM6208 (GCF_000702585) and *E. aurantiacum* NCTC13163 (GCF_900450545) were selected and downloaded from NCBI on 14 July 2021.

As *E. mexicanum* A-EM is a genome of phylogenetically closest type strain with available genome sequence and some genome including *E. mexicanum* HUD, *E. aurantiacum* DSM 6208 and *E. aurantiacum* NCTC13163 are also well-characterized. So retrieved all the genomes sequences and download from NCBI database and their comparative genome features with *E. aurantiacum* SW-20 are all listed in Table 1. The comparison and annotation of orthologous gene clusters were carried out by using OrthoVenn2 (https://orthovenn2.bioinfotoolkits.net/home, accessed on 22 June 2021) [31]. Sibelia software [32] was used and alignment of syntenic blocks were visualized in CGview [33] for genome synteny analyses. Predictions of KEGG pathways were carried out in GhostZ using the KEGG automatic annotation server (KAAS) [25] and the bi-directional best hit method.

### 2.6. Nucleotide Sequence Accession Number

The genome sequence of *E. aurantiacum* SW-20 has been deposited at NCBI under the accession number JAHXHC000000000 (BioProject ID: PRJNA748849 and BioSample ID: SAMN20346000). The RNA-seq data reported in this paper have been deposited at NCBI SRA database (SRR17327359 and SRR17327360). The strain *E. aurantiacum* SW-20 was cryopreserved in the laboratory of Nankai university.

## 3. Results and Discussion

### 3.1. Phylogeny of E. aurantiacum SW-20

The 16S rRNA gene sequence of *E. aurantiacum* SW-20 was found to be highly similar to the closest type strain *E. mexicanum* A-EM, which was isolated from seafloor hydrothermal vents (Caifan field, 14.0S 14.4 W) [34] and shared phylogenetic similarity with the other members of *Exiguobacterium* genus (Figure 1). The genomic length of *E. aurantiacum* SW-20 is 2,953,062 bp, with an average GC% content of 52.19%. The largest liner contig size is 2,044,816 bp, as shown in Figure 2, and the visualization is limited to the largest linear contig of the genome because these key genes that we care about can be found in this sphere. A total of 2979 CDS were predicted in the genome using Prokka v.1.

### 3.2. Characteristics of E. aurantiacum SW-20

To better understand the characteristics of *E. aurantiacum* SW-20, various experiments were conducted. As shown in Figure 3A, the colonies formed in the LB medium with 5% NaCl are mostly round and light yellow, with a smooth surface. The morphological characteristics of SW-20 were revealed by the SEM at a magnification of 20000×. It can be seen that the shape of bacteria was spherical in the 0% NaCl condition (Figure 3B), but when NaCl increased by 8%, the shape of the bacteria changed from a ball to a rod (Figure 3C). It can be seen that salinity has a certain influence on the morphology of SW-20. The growth curve of this strain under different salt conditions is shown in Figure 3D, which shows that the strain can grow under the condition of 10% NaCl. After 30 h of culture, the strain performance was better at 6% salinity than at 0% salinity. When the culture time was increased to 66 h, the strain demonstrated good growth potential at 8% salinity. As *E. aurantiacum* SW-20 was isolated from the oilfield, we wanted to see if this strain could degrade petroleum hydrocarbons. At the logarithmic growth stage, the strain was inoculated into crude oil degradation medium with a 1% inoculum amount and incubated for 7 days in a 30 °C constant temperature shaker. The method [3] was used to determine of n-alkanes content of crude oil before and after explanation. As shown in Figure 3E, the strain could degrade n-alkanes of all components of crude oil to a certain extent, with a particular emphasis on short and medium alkanes (C13–C26). In general, phenotypes are linked to genes, which in turn determine the function of bacteria. As our findings show that SW-20 has a certain salt tolerance and can degrade n-alkanes in crude oil, it is critical to further investigate the genes which may exist in the strain that are related to salt tolerance and petroleum hydrocarbon degradation.

### 3.3. Alkane Degrading Gene-Cytochrome P450 Gene

According to the genome, the cytochrome P450 gene from *E. aurantiacum* SW-20 was encoded and involved in alkane degradation. As P450 are not found in the largest contig but exist in contig 3, which is 132,548 bp long, and the cytochrome P450 genes involved in the metabolization of short or medium-chain alkanes i.e., C5–C11 were identified [35]. When cultured in basal mineral medium broth supplemented with 0.1% (*w*/*v*) hexadecane as the sole carbon source, strain SW-20 showed good emulsification and growth, confirming its ability for alkane degradation and metabolic utilization. What’s more, we detected the presence of a gene known as the haloacid dehalogenase type II, which is important in the degradation of chlorocyclohexane, chlorobenzene, chloroalkane and chloroalkene [36]. As the P450 cytochromes from bacteria are soluble heme-thiolate proteins that may mediate the terminal hydroxylation of alkanes [37]. They usually act as the terminal monooxygenases and frequently require the participation of two electron donors such as a FAD-containing NADH-dependent reductase and a ferredoxin [2Fe-2S], in to perform their function [38]. These transfer electrons from NADH to the heme domain of the P450 protein, resulting in O_2_ cleavage and substrate hydroxylation [39]. The cytochromes P450 could not only oxidize n-alkanes (usually range from C8 to C16) but also cycloalkanes and alkyl-aromatics [35,40], implying that strain SW-20 may play a putative role in degradation of both aliphatic and aromatic hydrocarbons in deep sea hydrothermal or reservoir environments.

Long-chain aliphatic and aromatic hydrocarbons are hydrophobic and insoluble in water, whereas biosurfactants, including glycolipids, lipopeptides, lipoproteins, phospholipids and fatty acids, can emulsify and solubilize hydrocarbons to promote their bioavailability [41]. From the lipid metabolism we can see that genes involved in lipid synthesis were discovered in the genome of *E. aurantiacum* SW-20, including CDP-diacylglycerol-serine O-phosphatidyl transferase gene (*pssA*), FAD-dependent oxidoreductase gene (*glpA*), 1-acyl-sn-glycerol-3-phosphate acyltransferase (*plsC*) and glycerol-3-phosphate acyltransferase (*plsY*). As fatty acids could enhance membranes fluidization of and adaptation to low temperatures or starvation in the deep sea, they can also enhance the aqueous solubility of hydrocarbon substrates in the reservoir environment [42]. In the genome of strain SW-20, genes involved in fatty acid biosynthesis including fatty acid dehydrogenase, transaminase and synthase were discovered, as well as one glycine cleavage system H protein (*gcvH*). Genes involved in alkane degradation such as cyclohexanone monooxygenase (cytochromes P450) were also found, but no *alkB*-related genes were discovered. Previous research has shown that some species from *Exiguobacterium* have the ability to degrade petroleum hydrocarbons [43,44,45], implying that strain SW-20 may have used the cytochromes P450 gene to enhance its hydrocarbon degradation ability [46].

### 3.4. Salt Tolerance Genes and Arsenic Tolerance Genes

In addition, the genome of *E. aurantiacum* SW-20 also contains some Na^+^/H^+^ antiporters, like the *nhaH*/*nhaC* gene. As *nhaC* has two copies, they are distributed on contigs 2 and 5 (79,395 bp) respectively. Previous studies have shown that *nhaH* genes can maintain the cells permeability of *E. coli* KNABC by expelling sodium ions and infiltration, both of which play important roles in the process of salt tolerance [47,48]. When cloned and with heterologous expression in vitro, Yang et al. discovered that the gene of Na^+^/H^+^ *nhaH* transporter from *Halobacillus dabanensis* D-8T (halophilic bacterium of bacillus) could enhance the salt tolerance of the strain’s action of force [49]. Some genes related to salt tolerance, such as the Na^+^/H^+^ antiporter and Na^+^/H^+^ transporter, can be found in the genome of *E. aurantiacum* SW-20. Some research has shown that amino acids like arginase or arginine can affect the ability of salt tolerance [50,51]. Experiments have shown that the addition of arginine could significantly improve the salt tolerance [52]. When fine glutamic acid was first synthesized in the cell of *H. elongata*, 3.4 M NaCl solution was used, and alanine and glutamine were synthesized when amino acids accumulated to a certain extent [53]. We can also find some arginine, glutamine and glutamic acid in our genome. In fact, organisms frequently have more than one way of dealing with extreme environments such as salt stress. In other words, when faced with extreme environments, organisms typically collaborate through multiple salt-tolerant mechanisms. In addition to the effects of transporters and amino acids, some extracellular polysaccharides as compatible substances also play an important role in regulating salt stress.

Furthermore, it is worth mentioning that polysaccharides are also essential for maintaining the balance of osmotic pressure. Previous studies have found that *Propionibacterium freudenreichii* could accumulate a high concentration of trehalose in response to the high permeability pressure [54]. At the present stage, two methods of trehalose synthesis were found. The first is the trehalose-6-phosphate synthetase/phosphatase pathway (Trehalose-6-phosphate synthase/phosphatase, OtsA-OtsB), and the second is the trehalose synthase (TreS) pathway. Previous research showed that high osmotic pressure enhanced the trehalose synthesis pathway of OTsA-OTsB [55]. We also found 30 genes related to glycan biosynthesis and metabolism in our genome, like the alpha-galactosidase *gala*, peptidoglycan pentaglycine glycine transferase *femX*, mannosyl-glycoprotein endo-beta-N-acetylglucosaminidase and 200 genes related to carbohydrate metabolism, such as glucose-1-phosphate adenylyltransferase *glgC*, glycogen phosphorylase *glgP*, cyclomaltodextrinase/maltogenic alpha-amylase/neopullulanase *cd*, and 51 genes related to lipid metabolism (like long-chain acyl-CoA synthetase *fadD*, glycerol-3-phosphate dehydrogenase *glpA*), which provided good evidence for the salt tolerance of *E. aurantiacum* SW-20.

When in an environment of high salt concentration, plants usually protect themselves by producing some reactive oxygen species (ROS). As many enzymes can remove the reactive oxygen species in cells, the photosynthesis efficiency of Zn/Cu SOD transgenic plants under oxidative stress is significantly improved when compared with the control plants [56]. The NTGST/GPX genes, which encode glutathione peroxidase and glutathione S transferase are overexpressed in tobacco, and the salt tolerance improves significantly [57]. It can be seen that genes related to SOD, glutathione peroxidase and glutathione S transferase may have a certain protective effect on alleviating the coercion under extreme conditions. Some genes were encoded in the genome of *E. aurantiacum* SW-20, including glutathione peroxidase gene (*gpx*), S-(hydroxymethyl) glutathione dehydrogenase/alcohol dehydrogenase gene (*frmA*), organic hydroperoxide resistance protein (*osmC*), superoxide dismutase, alkyl hydroperoxide reductase subunit F (*ahpF*) and peroxide-responsive transcriptional repressor (*perR*). It reveals that these genes may be related to the adaptation to extreme environments, and their specific function and mechanism must be explored further.

Aside from salt tolerance genes, arsenic resistance genes including *arsC*, *arsB*, *hesB*, *arsA*, *arsD* and *arsR* were found in the genome of *E. aurantiacum* SW-20. Because arsenic is a naturally occurring element, the International Agency of Research on Cancer has classified it as a class I carcinogen [58]. Arsenic contamination is derived from both anthropogenic sources, such as industrial waste or agricultural practices, and natural sources, such as dissolution of arsenic-containing minerals [59]. An arsenic resistance operon was found in the genome of *E. aurantiacum* SW-20, which included arsenate reductase family protein (*arsC*), arsenic transporter (*arsB*), Fe-S cluster assembly protein *hesB*, arsenical pump-driving ATPase (*arsA*), arsenite efflux transporter metallochaperone *arsD* and winged helix-turn-helix transcriptional regulator (*arsR*). The arsenic operon’s minimal constituents are three genes: a single transcriptional unit *arsR*, which encodes the arsenic transcriptional repressor, a membrane-bound arsenite permease *arsB*, which exports arsenite, and a reductase *arsC*, which converts arsenate into arsenite [60,61,62]. In bacteria, more complex arsenic resistance operons have been described, with additional genes encoding an ATPase (*arsA*), an arsenite chaperon (*arsD*), an organoarsenical oxidase (*arsH*) and the arsenite methyltransferase (*arsM*) [62,63,64,65,66]. Overall, the arsenic gene operons found in *E. aurantiacum* SW-20 differ from the minimal constituents of the arsenic tolerance operons, and they also differ from the complex arsenic operons, indicting a high potential in arsenic resistance. It is also useful in elucidating the resistance mechanism of arsenate in bacteria, especially *E. aurantiacum* SW-20, and it shows promise in exploring the specific oxidative stress reaction by reducing the formation of arsenic and/or the intracellular arsenic complex.

### 3.5. Genome Assemblies, Comparative Genome and Genome Synteny Analyses

The genome harbors 50 tRNA and 6 rRNA genes, one of which contains 18 types of tRNA genes and one 16S rRNA, four 23S rRNA and one 5S rRNA respectively. Furthermore, 2444 of the predicted protein coding genes were assigned to the KEEG metabolic pathways (Table 1). Based on the available complete genome of *E. aurantiacum* SW-20, the genome size ranges from 2.0 Mb to 4.0 Mb, indicating considerable plasticity in the *Exiguobacterium* species. Furthermore, the GC% content of the previously reported species also varied from 51.13% to 52.79%. The genome of *E. aurantiacum* SW-20 had a slightly higher GC% content (62.19%) and was closest to *E. mexicanum* A-EM (62.06%; Table 1).

A high GC% content may result in the adaptability to a complex environmental setting, as deep-sea hydrothermal environments and pan-genome profiles of bacteria shed light on their core and accessory genes with differences in genomic signatures. A comparison of predicted protein-coding genes for their orthologous relationship revealed that significant gene overlaps between *E. aurantiacum* and *E. mexicanum* genomes.

At the level of protein sequence, Veen analysis (Figure 4A) revealed that the three strains including *E. aurantiacum* SW-20, *E. mexicanum* A-EM and *E. mexicanum* HUD form 2877 clusters, 2857 orthologous clusters (at least contains two species) and 20 single-copy gene clusters. There are 2414 ortholog clusters shared by all three species, with 443 clusters shared by at least two genomes. A total of 20 gene clusters were unique to a single genome, with one belonging to *E. aurantiacum* SW-20 (Figure 4A). The genomes of *E. aurantiacum* SW-20 and *E. mexicanum* A-EM shared only 55 orthologous clusters (Figure 4A). The Venn diagram was used to compare *E. aurantiacum* SW-20 to *E. aurantiacum* DSM 6208 and *E. aurantiacum* NCTC13163. As we can see from Figure 4B, there are 2985 clusters, 2976 orthologous clusters (at least two species) and nine single-copy gene clusters. There are 2600 ortholog clusters shared by all three species, with 376 clusters shared by at least two genomes. A total of nine gene clusters were unique to a single genome, with five belonging to *E. aurantiacum* SW-20 (Figure 4B). The genomes of *E. aurantiacum* SW-20 and *E. aurantiacum* NCTC13163 shared six orthologous clusters exclusively, whereas *E. aurantiacum* DSM 6208 had no gene clusters, indicating that it had only one genome (Figure 4B).

A comparison of multiple alignment blocks between *E. aurantiacum* SW-20, *E. mexicanum* A-EM and *E. mexicanum* HUD revealed 10 syntenic regions distributed in the main scaffold (the largest scaffold representing for 69.24% of the whole genome) (Figure 4C). The highest level of synteny was observed between *E. aurantiacum* SW-20 and *E. mexicanum* HUD which shared 98.06% and 97.22% of total syntenic regions, respectively (Figure 4C). Comparison of multiple alignment blocks between *E. aurantiacum* SW-20, *E. aurantiacum* DSM 6208 and *E. aurantiacum* NCTC13163 revealed 43 syntenic regions distributed in the main scaffold (Figure 4D). The observed syntenies among *E. aurantiacum* SW-20, DSM6208 and NCTC13163 were low, accounting for only 5.81% of total syntenic regions (Figure 4D). The majority of metabolic pathways were found to be involved in the carbohydrate metabolism and the amino acid metabolism, according to a KEGG pathway enrichment analysis (Figure 4E). The number of genes involved in ‘Carbohydrate metabolism’ was greater in *E. mexicanum* HUD (240) and *E. aurantiacum* SW-20 (238) than in *E. mexicanum* AEM (214), *E. aurantiacum* DSM6208 (223) and *E. aurantiacum* NCTC13163 (220) (Figure 4E).

The genomes of these five strains were further screened for genes related to petroleum hydrocarbon degradation. The main gene clusters related to this phenotype are cytochrome P450 hydroxylase [35]. We found that these fives strains all encode for the same key enzymes i.e., cytochrome P450 hydroxylase. However, the accessory genetic components of each cluster differed across the five genomes (Figure 5B). In each of the five genomes, the genomic context surrounding cytochrome P450 hydroxylase differs (Figure 5B). When compared to others, the gene cluster in *E. aurantiacum* SW-20 appears to be more unique. For example, there are no IS200/IS605 family transposase, IS200/IS605 family transposase accessory protein and ferrous iron transport protein A when compared with others. In general, cytochrome P450s of bacterial requires ferrous iron transport protein for electron transfer. Increased enzymatic activities were observed when P450s were combined with other reductase domains [67].

Additionally, we can see that all of the genomes contained an arsenic resistance operon and a Na^+^/H^+^ antiporter protein gene (*nhaC*) cluster (Figure 5A,C), suggesting that this genus has the potential to be arsenic tolerant and salt resistant. The other four strains, with the exception of *E. mexicanum* A-EM, had identical genetic makeup including six genes (Figure 5A): transcriptional regulator *arsR*, arsenite efflux transporter metallochaperone *arsD*, arsenical pump-driving ATPase, Fe-S cluster assembly protein *hesB*, arsenic transporter *arsB,* and FAD-dependent oxidoreductase, which showed good application prospects in the field of pollution prevention and remediation of heavy metal arsenic.

We also compared the *nhaC*, which is associated with the salt tolerance. As shown in Figure 5C, *E. aurantiacum* SW-20, *E. mexicanum* A-EM and *E. mexicanum* HUD all have the same set of genes. The only difference is that the genetic compositions of *E. aurantiacum* DSM 6208 and *E. aurantiacum* NCTC13163 are identical, but they all lack MFS transporter. Furthermore, microbes frequently use dihydroxylation as an initial step in the aerobic degradation of aromatic compounds, and it is also used in the construction of recombinant pathways for degradation of halogenated aromatic xenobiotics, which provides an added advantage in the removal of halogen prior to oxidation of the aromatic ring.

### 3.6. Comparative Transcriptome Analysis of Genes Associated with Salt Tolerance

In order to further study the expression of salt-tolerant genes in the genome under different salinities, the transcriptomes of strains exposed to 0% and 8% NaCl conditions were tested, and a comparative transcriptome analysis was performed. The total number of genes was 2934 at 0% and 8% salinity with 9 (0% NaCl) and 8 (8% NaCl) unique genes, respectively.

Transcriptome analysis revealed that genes associated with salt tolerance were up-regulated to some extent as salinity increased. Some genes related to salt-tolerance were listed in Table 2 when the fold change (FC) (8%/0%) was ≥5. We can see that the gene *yuiF*, which is related to the Na^+^/H^+^ antiporter, has significantly increased. Furthermore, *arsA*, *arsR* and *arsD* all show significant increases. Because the three genes belong to the arsenical pump-driving ATPase, winged helix-turn-helix transcriptional regulator and arsenite efflux transporter metallochaperone operons, it is possible that these genes related to the arsenic operon may play a role in tolerance to extreme environments. Besides, some genes related to cytochrome P450, Fe-S cluster assembly protein, glycerol dehydrogenase and VOC family protein were significantly up-regulated, indicating good research potential for dealing with extreme environments and even for the pollution treatment of heavy metals in arsenate.

Thus, our comparative genomic study revealed significant differences in genomic arrangement and functional attributes involved in hydrocarbon degradation among the hydrocarbonclasticus members of *E. aurantiacum* SW-20. Future functional studies involving transcriptional profiling of *E. aurantiacum* SW-20 under a wide range of hydrocarbon-relevant conditions will greatly expand our insights into marine hydrocarbon degradation in a reservoir environment. Our study contributes to the knowledge on the genetics of petroleum hydrocarbon clastic members of *Exiguobacterium* and provides insight into the characteristics of *E. aurantiacum* SW-20 that may be instrumental in tolerating and thriving in the extreme environment of a reservoir. Overall, our research indicated that *E. aurantiacum* SW-20 could be used in the degradation of certain types of petroleum hydrocarbons, providing potential and improved application prospects for the treatment of petroleum hydrocarbon pollutants and even bioremediation.

## 4. Conclusions

Previous studies have paid less attention to the differences in the composition of gene clusters, which are related to salt tolerance, arsenic tolerance and hydrocarbon decomposition. By comparing the genes of five species at different genus levels, it was expected to reveal the influence of different gene cluster compositions in the genus of *Exiguobacterium* and provide guidance for further exploration of the relationship between genes composition and their functions in extreme conditions. In fact, for an extreme environment, screening extreme environment microbes is necessary and important. With the development of the petrochemical industry, it is urgent to screen bacteria with high-efficiency salt-tolerant as well as petroleum hydrocarbon bioremediation. 

Genes in *E. aurantiacum* SW-20 were analyzed at the genomic and transcriptional levels, and our research showed that *Exiguobacterium* has good potential not only in high salt conditions but also in petroleum hydrocarbon degradation. Besides, an arsenic tolerance operon was discovered that differed not only in the minimal constituents of the arsenic tolerance operon but also in the complex arsenic operons, indicating that it has a high potential in arsenic tolerance even in the field of heavy metal pollution remediation. Furthermore, by comparing the composition of the same gene cluster, we can find some specific differences which will serve as a reference for further research into the function of these genes as well as their mechanism in extreme conditions including salt-tolerant, heavy metal tolerance, and petroleum hydrocarbon degradation.

## Figures and Tables

**Figure 1 microorganisms-10-00066-f001:**
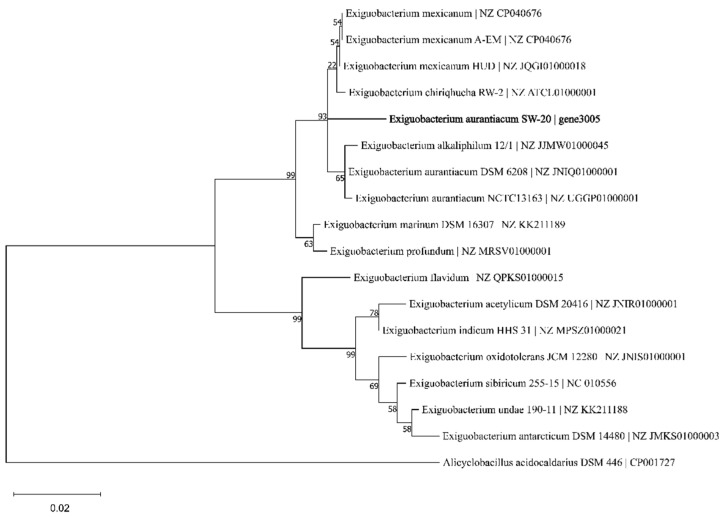
Phylogenetic tree of *E. aurantiacum* SW-20 and its closest related species based on 16S rRNA gene sequence similarity using *Maximum-Likelihood* algorithm (*Jukes-Cantor* model). The outgroup used was *Alicyclobaciillus acidocaldarius* DSM 446. Bootstrap value represents 2000 replicates and random seeding. Tree was built using Mega v.11.

**Figure 2 microorganisms-10-00066-f002:**
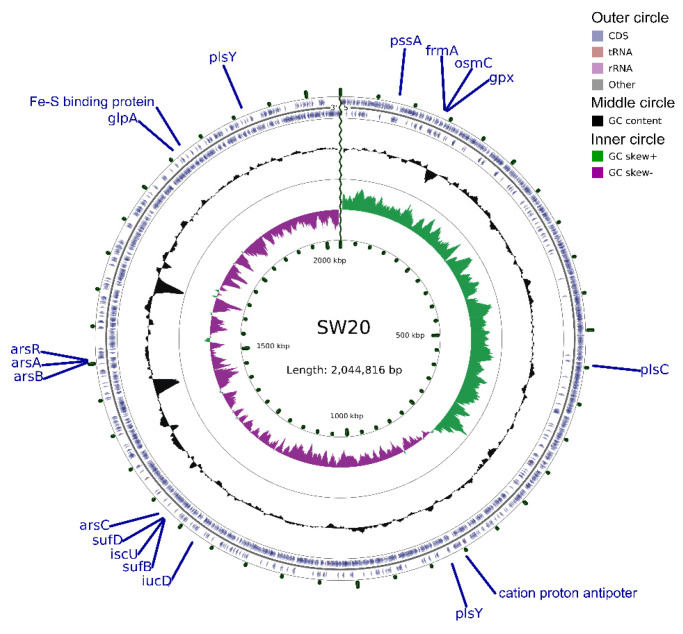
Circular representation of genome and features of *E. aurantiacum* SW-20. The contents of the featured rings (starting with the outermost ring to the center) are as follows. From the outside to the inside, outer circle contains CDS, tRNA and rRNA on the positive and negative strands respectively. The middle circle indicates that the GC% content in this region is higher than the average GC% content of the whole genome, and the higher the peak value is, the greater the difference between the region and the average GC% content is. The inward part indicates that the GC% content in this region is lower than the average GC% content of the whole genome, and the higher the peak value is, the greater the difference between the region and the average GC% content is. The inner circle is GC-skew value and the specific algorithm is G − C/G + C, which can assist in determining the leading chain and the lagging chain. Generally, the leading chain GC Skew > 0, while the lagging chain GC Skew < 0, which can also assist in determining the starting point of replication (the minimum cumulative offset) and the end point (the maximum cumulative offset), especially for the circular genome. The innermost ring means the genome size marker.

**Figure 3 microorganisms-10-00066-f003:**
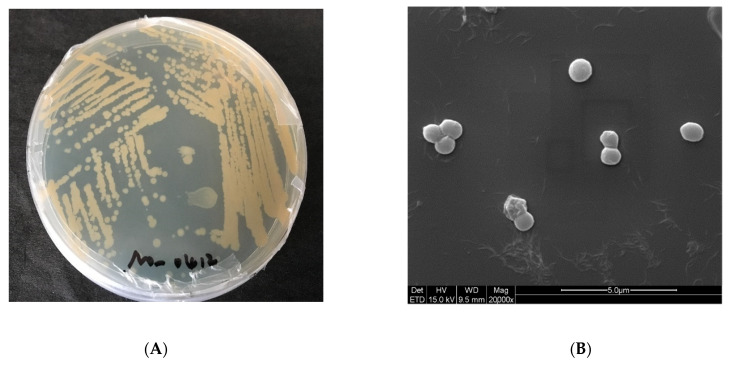
Characteristics of *E. aurantiacum* SW-20 (**A**) Colony morphology of the strain in 5% LB solid medium (**B**) SEM imagines at 0% NaCl (**C**) SEM imagines at 8% NaCl (**D**) Growth curve on different content of NaCl (**E**) Degradation diagram of n-alkanes in crude oil.

**Figure 4 microorganisms-10-00066-f004:**
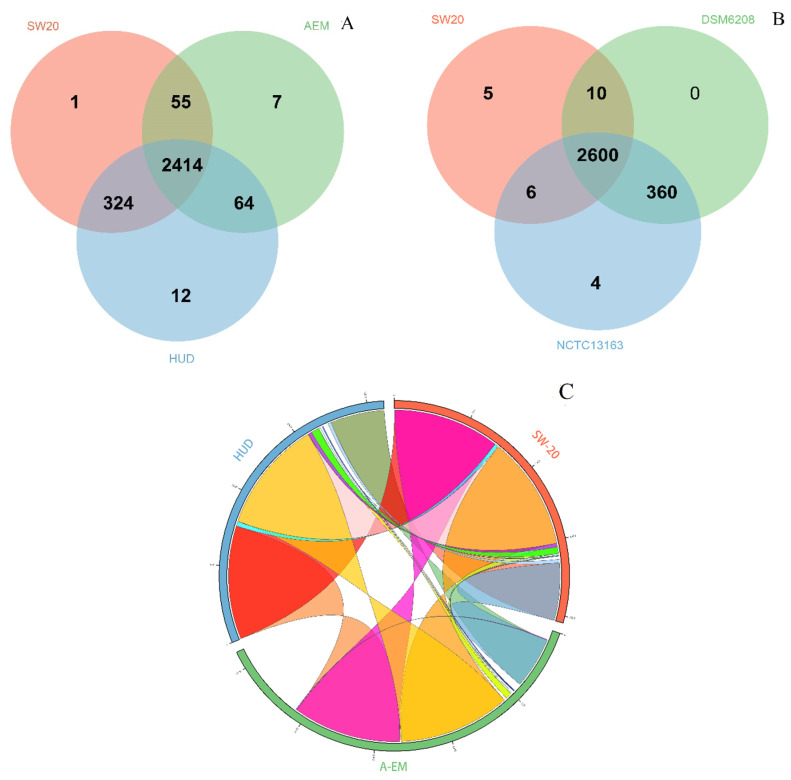
Analysis of orthologous genes and genomic synteny in *E. aurantiacum* SW-20, *E. aurantiacum* DSM 6208, *E. aurantiacum* NCTC13163, *E. mexicanum* A-EM and *E. mexicanum* HUD (**A**) Venn diagram represents distribution of shared and unique gene clusters among SW-20, A-EM and HUD (**B**) Venn diagram represents distribution of shared and unique gene clusters among SW-20, DSM 6208 and NCTC13163 (**C**) Genomic synteny shared between SW-20, A-EM and HUD. (For interpretation of the references to color in this figure legend, the reader is referred to the web version of this article.) (**D**) Genomic synteny shared between SW-20, DSM 6208 and NCTC13163. (For interpretation of the references to color in this figure legend, the reader is referred to the web version of this article.) (**E**) KEGG pathway enrichment analysis represented through bar chart shows distribution of number of proteins in 18 different KEGG metabolic function categories annotated using the KAAS.

**Figure 5 microorganisms-10-00066-f005:**
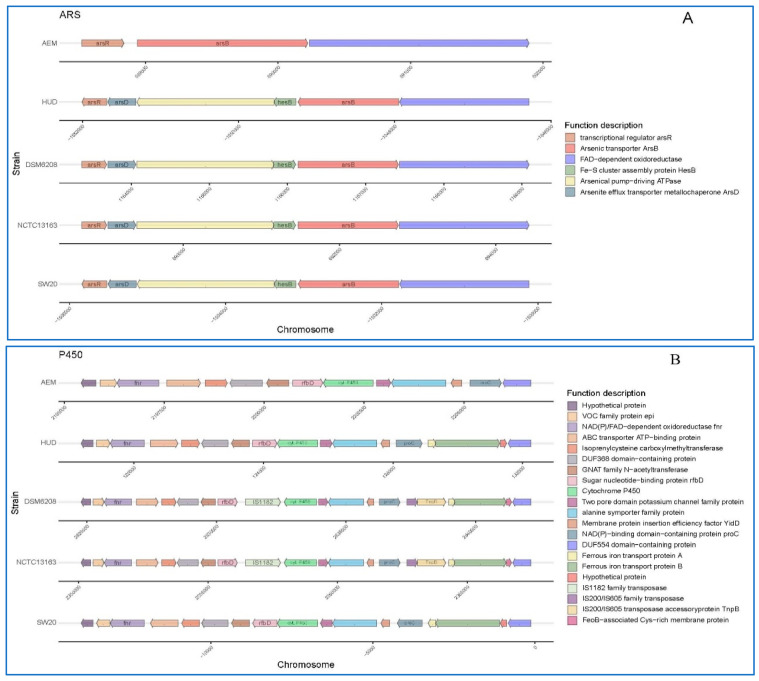
Schematic comparison of the three main gene clusters for different function based on annotations in the KEEG database: (**A**) Arsenic tolerance (**B**) Cytochrome P450 hydroxylase and (**C**) Na^+^/H^+^ antiporter protein in *E. aurantiacum* SW-20, *E. aurantiacum* DSM 6208, *E. aurantiacum* NCTC13163, *E. mexicanum* A-EM and *E. mexicanum* HUD.

**Table 1 microorganisms-10-00066-t001:** Genome features of *E. aurantiacum* SW-20, *E. aurantiacum* DSM 6208, *E. aurantiacum* NCTC13163, E. mexicanum A-EM and E. mexicanum HUD.

Genome Features	*E. aurantiacum* SW-20	*E. aurantiacum* DSM 6208	*E. aurantiacum* NCTC13163	*E. mexicanum* A-EM	*E. mexicanum* HUD
Genome size (bp)	2,953,062	3,037,601	3,125,661	2,693,465	3,359,295
GC%	52.19	52.79	52.58	52.06	51.13
Contigs	69	2	4	4	825
Scaffold N50 (bp)	2,044,816	2,911,818	2,911,806	2,615,862	2,107,289
Total numbers of CDS	2979	2985	3068	2690	3621
tRNA	50	67	67	40	69
rRNA (5S, 16S, 23S)	1, 1, 4	10, 9, 9	10, 9, 9	4, 4, 4	4, 2, 3
KEGG metabolic path-ways	2444	2387	2378	2243	2515

**Table 2 microorganisms-10-00066-t002:** Transcriptome upregulates genes and their annotations when FC (8%/0%) ≥5.

Genes	Descriptions	FC (8%/0%)
*yuiF*	Na^+^/H^+^ antiporter	6.596
*arsA*	arsenical pump-driving ATPase	5.268
*arsR*	winged helix-turn-helix transcriptional regulator	13.349
*arsD*	arsenite efflux transporter metallochaperone *ArsD*	6.317
*gldA*	glycerol dehydrogenase	8.733
*galK*	galactokinase	6.482
*hisD*	histidinol dehydrogenase	5.557
*HesB*	Fe-S cluster assembly protein	8.168
-	cytochrome P450	5.819
-	ABC transporter permease	7.188
-	VOC family protein	7.931
*ahpC*	MULTISPECIES: peroxiredoxin	5.167
*phhA*	aromatic amino acid hydroxylase	11.296
*frmA*	glutathione-dependent formaldehyde dehydrogenase	6.442
*gnd*	decarboxylating 6-phosphogluconate dehydrogenase	1221.275

- means that the name of these genes is not clear.

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
