# Peer review of "Genetic and Comparative Genome Analysis of Exiguobacterium aurantiacum SW-20, a Petroleum-Degrading Bacteria with Salt Tolerance and Heavy Metal-Tolerance Isolated from Produced Water of Changqing Oilfield, China"

_microorganisms, 2021, doi:10.3390/microorganisms10010066_

Round 1
Reviewer 1 Report
The authors have done a good job revising the document. I don't have further comments to make.
Author Response
Response: First of all, thank you for taking time out of your busy schedule to carry out a rigorous review of our manuscripts. Secondly, thank you for your recognition and affirmation of our work, we will make persistent efforts to do better. Finally, thank you again and best wishes for you.
Reviewer 2 Report
This study consists of genome sequencing, transcriptomics, salt tolerance and oil degradation studies of the bacterium E. aurantiacum strain SW-20. The manuscript is generally organized well, however, poor grammar throughout complicates readers’ understanding and English should be extensively revised prior to publication. The following scientific and formatting points should also be addressed:
- What about C5-C12 alkanes? Why were these not shown in fig 3E?
- Growth curves should be referenced or shown for grown in crude oil medium, because this reviewer is not convinced by the small changes in C13-C35 alkanes in fig 3E. Was any effort carried out to characterize the putative products of alkane metabolism?
- Lines 356-357 typos in GC content %
- Will these sequencing data be available for other scientists? If so how?
- Please comment on the >1000 fold increase seen for gnd in 8% vs 0% NaCl
- Methods do not include formulation of LB medium. Please include because different recipes have different NaCl content. For the 0% experiments, was any NaCl added? This is not clear from the manuscript.
- Does the organism have any antibiotic tolerances?
Author Response
Point 1: What about C5-C12 alkanes? Why were these not shown in fig 3E?
Response: Thank you for your advice. At present, we paid more attention to the n-alkanes after C12, so during the initial setup of the program, it starts from C13. As n-hexane was used as an organic phase to extract crude oil in the determination medium and itself is C6, Therefore, this program can ensure the peaks of n-alkanes after C12 and do not overlap with the n-alkanes and become unrecognizable. Therefore, in order to ensure the accuracy of experimental data, we only counted the degradation efficiency of n-alkanes after C12. Thank you again for your good questions.
Point 2: Growth curves should be referenced or shown for grown in crude oil medium, because this reviewer is not convinced by the small changes in C13-C35 alkanes in fig 3E. Was any effort carried out to characterize the putative products of alkane metabolism?
Response: Thank you for your advice. Growth curves have been referenced in manuscript.
To better explore the concrete putative products of alkane metabolism, we have already got some gene clusters mentioned, and our next work is to explore more characterizes and their functions and research how they work by heterogeneous expression.
Point 3: Lines 356-357 typos in GC content %
Response: Thank you for your advice. We have revised it in the new manuscript as “GC% content”.
Point 4: Will these sequencing data be available for other scientists? If so how?
Response: Thank you for your question, the sequence data of our genome has already submitted to the NCBI database (accession No. JAHXHC000000000). As we are currently conducting further research, the data may not be released until this part of the experiment is over, so it will be released once those work is finished.
Point 5: Please comment on the >1000 folds increase seen for gnd in 8% vs 0% NaCl
Response: Thank you for your question. In our transcriptome, as the value of 0% NaCl and 8% NaCl is 1.84 and 2513.68 respectively, we can see that the low denominator value 1.84 makes a high fold for 8% vs 0% NaCl for gnd.
Point 6: Methods do not include formulation of LB medium. Please include because different recipes have different NaCl content. For the 0% experiments, was any NaCl added? This is not clear from the manuscript.
Response: Thank you for your good advice and it’s really a good suggestion. We have revised this manuscript and included the formulation of LB medium clearly in the Methods. In our experiment, For the 0% experiments, it means that only use LB medium but not add extra NaCl. For the 8% experiments, it means that based on the LB medium, add an additional 8% of NaCl. We have modified it more clear in our manuscript. Thank you again for your advice.
Point 7: Does the organism have any antibiotic tolerances?
Response: Thank you for your good question. To explore whether the organism have any antibiotic tolerances, a basic experiment was carried out. First, two common tolerances were selected, including amoxycillin sodium and kanamycin sulfate. By adding two kinds of antibiotics with different contents in LB medium respectively and taking no antibiotics as the control. Then measured the OD600 after the bacteria cultured in 30℃ shaking table for 24 h inoculation. The results showed that this organism could not tolerance amoxycillin sodium and could tolerance 20 μg/mL (OD600:3.54) kanamycin sulfate (CK OD600:3.88). In other words, this strain was somewhat resistant to ampicillin sodium when compared to kanamycin sulfate. In fact, study on tolerance of organism is really an interesting part for further research, and we'll probably do some work on that in the future. Thank you again for such a good question.
Besides, based on the pathogenic system part of our genome data analysis, it can be seen from the drug resistance gene prediction plate that this organism has the following related genes, including: Macrolide antibiotic; Tetracycline antibiotic; Fluoroquinolone antibiotic; Glycopeptide antibiotic and et al, which suggested that bacteria may also have other resistance, which needs further study.
In order to make the English language more formal, we looked for some English majors to modify our manuscript, hoping that it could meet the requirements of journal.
First of all, thank you for taking time out of your busy schedule to carry out a rigorous review of our manuscripts. Secondly, thank you for your recognition and affirmation of our work, we will make persistent efforts to do better. Finally, thank you again and best wishes for you.

This manuscript is a resubmission of an earlier submission. The following is a list of the peer review reports and author responses from that submission.
Round 1
Reviewer 1 Report
Generally, the manuscript is well written, the introduction is sufficient, and the methods are well described and justified. the manuscript highlighted some novel results which might be helpful for further study of adaptation strategies of E. aurantiacum SW-20. The discussion is a little lengthy but comprehensive. I have some minor comments below:
1. Check the written format. Some paragraph remains bold and large fond size (2.3. Comparative genome and genome synteny analyses, ….).
2. "Samples of stratum water from the reservoir environment of Changqing Oilfields (China) were collected and E. aurantiacum SW-20 was isolated by using Luria-Bertani solid medium by adding lower content of NaCl. " Authors needs explain in details how did they specifically isolate E. aurantiacum from stratum water in basic medium just Luria-Bertani . Because, there will be other different types of bacterial population. Either they have done any primary characterization then they have to explain and write.
3. The genome of E. aurantiacum SW-20 has been compared with other four strains but there is nothing mentioned in abstract or conclusion. What is the purpose of comparison.
Reviewer 2 Report
# Minor
1) Title: Check the spelling (Prduction Water). Should it be produced water?
2) Section 2: The authors should provide versions of tools used in the article.
3) Section 2: Prokka should be mentioned in the material and method section as it was used to make a Figure 2.
4) Section 2.2: The authors should provide the date that the genomes were downloaded from NCBI.
5) Section 2.3: Would it be possible to show a maximum likelihood tree for the compared genomes? The Neighbor Joining is clustering algorithms that can make quick trees but is not the most reliable, especially when dealing with closely related genomes.
6) Figure 2: Could the authors justify the reason to visualize only the largest linear contig of the genome? This visualization could make audiences a confusion since the genome size of the genome is 2.9 Mb not 2.0 Mb.
7) Figure 2: The authors should state in the text of Figure 2 that the visualization is for only the largest linear contig of the genome.
8) Figure 2: Check the cropped text at the figure (cation proton ant...)
9) Figure 2: The color keys on the legend should be divided based on each ring.
10) Table 1: Check the cropped text at the last column (E. mexicanum … HUD)
11) Figure 4: The authors should use the brighter color for the CDS so audiences can see the text clearer.
12) Figure 4: The authors should make the text bigger so audiences can see the text clearer.
13) Section 3.4: Check the consistency of the font size.
# Major
1) Missing a conclusion part of the article.